# How Does Water-Stressed Corn Respond to Potassium Nutrition? A Shoot-Root Scale Approach Study under Controlled Conditions

**Lionel Jordan-Meille [1,\*], Elsa Martineau [1], Yoran Bornot [2], José Lavres [3], Cassio Hamilton Abreu-Junior [3] and Jean-Christophe Domec [1]**

[1] ISPA, Bordeaux Sciences Agro, UMR 1391, INRA, 33140 Villenave d'Ornon, France; elsa-martineau@hotmail.fr (E.M.); jc.domec@agro-bordeaux.fr (J.-C.D.)
[2] INRA UMR 1137 Ecologie et Ecophysiologie Forestières, 54280 Champenoux, France; bornot.yoran@outlook.fr
[3] USP-CENA, Plant Nutrition Laboratory, University of Sao Paulo, Piracicaba, SP 13416-000, Brazil; jlavres@cena.usp.br (J.L.); cahabreu@cena.usp.br (C.H.A.-J.)
\* Correspondence: lionel.jordan-meille@agro-bordeaux.fr; Tel.: +33-557-350-758

**Abstract:** Potassium (K) is generally considered as being closely linked to plant water dynamics. Consequently, reinforcing K nutrition, which theoretically favors root growth and specific surface, extends leaf lifespan, and regulates stomatal functioning, is often used to tackle water stress. We designed a greenhouse pot-scale device to test these interactions on corn (*Zea mays* L.), and to analyze their links to plant transpiration. Three levels of K nutrition were combined with two water-supply treatments. Shoot and root development and growth were continuously measured during a 60-day-long experiment. Individual plant transpiration was measured by weighing pots and by calculating water mass balances. The results showed that, although K deficiency symptoms resembled those caused by water shortage, there was no advantage to over-fertilizing water-stressed plants. K failed to decrease either the transpiration per unit leaf surface or to improve water use efficiency. The link between K nutrition and plant transpiration appears solely attributable to the effect of K on leaf area. We conclude that K over-fertilization could ultimately jeopardize crops by enhancing early-stage water transpiration to the detriment of later developmental stages.

**Keywords:** potassium supply; drought; *Zea mays* L.; root architecture; leaf area; transpiration

## 1. Introduction

Corn (*Zea mays* L.) is a crop that is very sensitive to water stress [1,2], which often makes irrigation necessary in most growing areas [3], such as in the southwest of France, a major corn production region. In this temperate part of Western Europe, annual rainfall is expected to decrease by the year 2050, coupled with an increase in the frequency and duration of summer droughts and temperature-driven evaporative demand [4,5]. Such climatic changes have highlighted the need to understand the key processes that can allow plants to acclimate to recurrent dry summers [6,7]. Fertilization with, in particular, the application of potassium (K), is one of the most commonly accepted strategies used to alleviate symptoms of water stress in several agricultural crops, including corn [2,8–10].

Plants' water stress decreases the leaf growth rate [11], partly because of a decrease in cell turgor [12], which ultimately leads to reduced leaf size and senescence [13]. The excess of sugar resulting from this limiting growth is sent to the roots, thus increasing the root-to-shoot biomass ratio [14,15]. Water stress also modifies root structure, with an increase in the root-specific area [16,17]. These responses, which promote better plant-resistance to soil water shortage, presumably depend on

K nutrition [18] or, at least, interact with it [19,20]. Three main reasons can explain the influence of K on plant-water status, which are as follows.

First, K is the ion that has the most influence on plant cell osmotic water potential which, in turn, controls cell- and leaf-growth [12,19,21]. Guard cells are known to be responsive to K, a major determinant of their functioning [22,23]. Experimental studies have long supported that K increases stomata regulation under water stress [24]. Hence, a deficient amount of K would prevent the stomata either opening or closing fully, even under severe water stress. Contradictory results have, however, been pointed out, depending on whether measurements were performed at a leaf, plant, or field scale [25]. Moreover, recent data clearly show that potassium's influence on transpiration is linked to mesophyll diffusion conductance more than to stomatal conductance [26]. Nevertheless, whatever the mechanism, a strong impact of K nutrition on water transpiration can be expected.

Second, K allows leaves to neutralize free radicals, known as "reactive oxygen species" [27,28]. These molecules accumulate during periods of water stress, causing protein and lipid peroxidation, and cell structure damage; their neutralization increases leaf lifespan. The influence of K on the persistence of green leaves has been identified [29].

Third, K facilitates the transformation in leaves of glucose into sucrose, thus favoring shoot-to-root sugar transport [20,30–32]. This preferential carbon allocation to roots impacts root morphology by enhancing primary and lateral root diameters [33] and by promoting root hair growth [34,35]. The role of K on root morphogenesis is not only due to its effect on carbohydrate mobility; it may possibly act on phytohormone activation [36], such as on abscisic acid production [37,38]. Hence, K can modify plant accessibility to water via its impact on root architecture.

In short, the literature provides several reasons for linking K nutrition to plant physiological acclimation to leaf water stress. However, knowledge gaps still exist concerning the combined effects of K and water stress on plant structure and whole-plant water consumption. This is especially true for corn, a fast-growing crop that requires a lot of water to reach maturity and yield a sustainable production, in a short space of time. The main objectives of the present experiment were to study the interactions between three K nutrition levels (K0, K1, K2) and two water-supply treatments (W+, W−) on corn shoots and below-ground development and growth, to determine whether K nutrition could limit and possibly mitigate the effects of water stress. We hypothesize that: (1) the effects of K deficiency and water shortage lead to comparable morphological root and shoot changes, as K shares some physiological functions with water; (2) K over-fertilization partially offsets the deleterious effects of drought; in which case; (3) this type of compensation is partly due to increases in leaf lifespan and in root specific area, together with a better transpiration regulation.

## 2. Materials and Methods

### 2.1. Plant Preparation

The experiment was carried out at the French National Institute for Agronomy (INRA, Bordeaux, France). In early March, corn seeds (*Zea mays* L.) were germinated in the dark at 21 °C on paper towels moistened with distilled water. After emergence of the first visible leaf (5 days), plants without any visual growth defects were selected and transplanted into 10-litre pots, filled with 14 kg of umbric ortsteinic podzol soil collected on a long-term potassium (K) fertilization trial [2]. The physico-chemical soil properties were as follows: 4.7% clay, 2.1% silt, 93.3% sand, 3.9% organic matter, with CEC 5 cmol $kg^{-1}$, a pH of 5.3, and a bulk density of 1.4. For the first two weeks after transplanting the germinated seeds, 150 mL (for a total of 2 L) of the following nutrient solution was added to each pot each day: $(NH_4)_2SO_4$ (72.6 mg $L^{-1}$), $Ca(NO_3)_2 \cdot 4H_2O$ (651.8 mg $L^{-1}$), $Mg(NO_3)_2 \cdot 6H_2O$ (87.18 mg $L^{-1}$), $MgSO_4$ (132 mg $L^{-1}$), $NaH_2PO_4$ (6.6 mg $L^{-1}$), $Na_2HPO_4 \cdot 12H_2O$ (19.7 mg $L^{-1}$), $MnSO_4 \cdot H_2O$ (0.615 mg $L^{-1}$), $ZnCl_2$ (0.21 mg $L^{-1}$), $CuSO_4 \cdot 5H_2O$ (0.047 mg $L^{-1}$), $H_3BO_3$ (0.562 mg $L^{-1}$), and $(NH_4)_6Mo_7O_{24} \cdot 4H_2O$ (0.322 mg $L^{-1}$).

## 2.2. Experimental Design: K Fertilization and Watering Modalities

To assess the effects of water, K, and their interaction on plant structural parameters, a randomized complete block design was set up in greenhouse conditions with two water treatments being combined with three K treatments. A total of 42 pots of corn were used (two water treatments, three K treatments, and seven replicates per treatment). The three K treatments, hereafter called K0, K1, and K2, were obtained by using soil fertilized with three different concentrations of K, corresponding respectively to 5, 17, and 30 mg kg$^{-1}$, expressed as exchangeable K by an ammonium acetate extraction [39].

During the first 37 days after leaf emergence (DAE), all the plants were well watered. This consisted in maintaining soil moisture at around 80% of the water-holding capacity (20% of volumetric soil water content) by weighing each pot between three and five times a week, according to the climate. In all, the water adjustment was performed 22 times; there were 17 weighings during the water treatment differentiation. To avoid affecting soil nutrient concentration, deionized water was used. After this delay, half of the plants continued to be well watered, according to the same protocol (W+ treatment), while the drought treatment (W−) was started on the other half of the pots; this consisted in maintaining soil moisture content near the wilting point (5% of volumetric soil water content). The water stress treatment was not applied immediately after transplanting the seedlings, in order to avoid plant mineral nutrient deficiency during early plant development due to a lack of water uptake. The calculation of the water amount required for each target (5% or 20% of volumetric soil water content) took into account the fresh biomass of the growing plants. These calculations were based on supplementary corn material, grown in the glasshouse and regularly cut and weighed. Moreover, three repetitions of bare soil pots filled to the two target water contents (20% and 5%) were weighed daily in order to measure the direct soil evaporation. Just before the destructive sample, pre-dawn water potentials were measured with a Scholander chamber. Five replicates per treatment were sampled, and each last ligulate leaf was carved in order to insert its main rib into the conical seal of the chamber. The first non-destructive measurements were performed at 22 DAE, and the last measurements (destructive samplings) were done at 65 DAE. The leaves of all 42 plants were numbered from the first true leaf (not cotyledon) to the top of the plant, so that the different K and water treatments could be compared at single development stages.

Thermal time (degree-days, dd) was calculated as the sum of cumulative differences between daily mean temperature and a base temperature (taken at 10 °C). Using this reference, water stress was then applied at 380 dd, and the experiment lasted until 700 dd. Greenhouse environmental data (temperature and photosynthetic active radiation) were recorded every 10 min using a datalogger (CR1000, Campbell Scientific, Logan, UT, USA).

## 2.3. Leaf Parameters and Leaf Area Calculations

The phyllochron, which corresponds to the interval between the sequential emergence of two successive leaves on the main stem, was determined from the linear relationship between the number of visible leaf and thermal time [40]. The calculation was performed between 380 and 700 dd, corresponding to the period during which water treatments differed. Individual leaf length (L), width (w), and growing-status (senescing, ligulate, or non-ligulate) were measured six times (including the final destructive measurements) throughout the whole experiment (approximately once a week). Leaf area (LA) was determined as [41]:

$$\text{LA (ligulate leaf, m}^2) = 0.75 \times \text{L (m)} \times \text{w (m)} \tag{1}$$

$$\text{LA (non-ligulate leaf, m}^2) = 0.5 \times \text{L (m)} \times \text{w (m)} \tag{2}$$

The plant green leaf area was calculated using the number of visible leaves, the number of senescent leaves, and the individual leaf surface area (Equations (1) and (2)). The relative senescent leaf values were calculated by dividing the area of the senesced leaves by the total leaf area. Specific

leaf area (SLA) was calculated as leaf area ($m^2$) divided by leaf dry mass (kg). Moreover, we paid particular attention to leaf 10 (taken as a common reference for all treatments), taking daily length measurements in order to calculate its elongation rates (LER in cm $dd^{-1}$). The calculation was based on the relationship between the successive length measurements during their quasi-linear elongation period and thermal time [41].

Daily values of green leaf areas were needed in order to parallel water transpiration data, we used a logistic curve fitted on the six measurement dates to make interpolations. In the special case of K0-W− plants, we used an exponential model to fit the evolution of their leaf area because those plants had not finished their growth by the end of the experiment (700 dd).

### 2.4. Continuous Shoot Fresh and Dry Matter Simulations

We simulated the continuous evolution of the shoot biomass, mirroring the changes in leaf area values. To do so, the model was calibrated with final fresh and dry biomass, measured during the destructive sampling (700 dd). The same specific leaf area was assumed for these calculations.

### 2.5. A Water Mass Balance to Calculate Soil Water Content and Transpiration

At any given time, pot volumetric soil water content (SWC) was calculated as:

$$SWC\ (\%) = (\ (Pot_d\ (kg) - DS\ (kg) - FM)/DS\ (kg)\ ) \times Dens \qquad (3)$$

with $Pot_d$: Mass of pot (kg) before watering; DS: Mass of pot with dry soil and without plant, equal to 14 kg; FM: Fresh mass of corn (kg) calculated as explained in Section 4.3; Dens: Soil bulk density, equal to 1.4.

The water transpired by each plant between two irrigation dates was calculated as:

$$Transp_{d-d\text{-}1}\ (L) = Pot_{d\text{-}1}\ (kg) - Pot_d\ (kg) + \Delta FM\ (kg) - Evap\ (kg) \qquad (4)$$

with $Pot_{d\text{-}1}$: Mass of pot (kg) just after re-adjustment to target humidity, during the previous irrigation date (date d-1); $Pot_d$: Mass of pot (kg) before watering at date d; $\Delta FM$: Increase in corn fresh matter (kg) between two irrigation dates (d and d-1), as explained in Section 4.4; Evap: Daily evaporation (kg) measured on bare pots for the corresponding water treatment between two irrigation dates (d and d-1).

The water transpiration was only calculated from 380 to 700 dd, corresponding to the period during which water treatments differed (17 measurement dates, 16 transpiration periods). This led to the calculation of the total amount of water transpired for each plant. Over the same period, we calculated, at pot scale, the water transpired per unit time (day) and per unit green leaf area ($m^2$). The grand mean of this transpiration per unit leaf area, and over the whole period (380–700 dd), was calculated as the mean of the 16 transpiration periods.

### 2.6. Long-Term Water Use Efficiency

Water use efficiency was calculated during each of the 16 transpiration periods after water treatments had differed. It was based on plant dry matter calculations (Section 4.4) and measurements of transpiration:

$$WUE\ (g\ L^{-1})_{d-d\text{-}1} = \Delta DM\ (g)/Transp\ (L)_{d-d\text{-}1} \qquad (5)$$

with $\Delta DM$: Increase of the dry matter of corn (g) between two irrigation dates (d and d-1), $Transp_{d-d\text{-}1}$: Water transpired (L) by each plant between two irrigation dates.

The grand mean of this WUE, and over the whole period (380–700 dd), was calculated as the mean of the 16 transpiration periods.

### 2.7. Root Measurements

The root system was separated by phytomers (P), which consist in a repetition of constructional units represented by successive horizontal circles from which primary roots emerge at the base of the stem (labeled from P1 at the bottom to P6 at the top [42]. In corn, the root system is composed of primary roots, carrying secondary roots (also called "laterals") growing below the apical zone. The roots were separated from the soil particles by dry-sieving them over a 2-mm wire mesh. The cleaned roots were submerged in 10% ethanol at 4 °C before being scanned. The root morphology was investigated using a root image analysis software (WinRHIZO Version 2005a, Regent Instruments Inc., Nepean, ON, Canada). The images were analyzed to identify the root morphological parameters. Roots with a diameter greater than 5 mm were excluded from the analysis, as only very few roots were above this value. Conversely, during cleaning, the smallest root hairs could not be retained, which meant that the finest root diameters exceeded 0.05 mm. The parameters scanned by the software were root length (cm), average diameter (mm), root volume (cm$^3$), number of root tips, and root surface area (cm$^2$). Specific root area (SRA in m$^2$ kg$^{-1}$) was calculated for each sample after drying the roots at 105 °C for 48 h as:

$$\text{SRA (m}^2 \text{ kg}^{-1}) = \text{Root area (m}^2)/\text{Dry matter (kg)} \tag{6}$$

The root morphology was described using the distribution of root surface by diameter class for root diameters comprised between 0.05 and 5 mm. The root image analysis software allowed us to calculate the cumulated surface, the "relative root area" and the "proportion of roots" corresponding to a given class diameter (i.e., corresponding to all those roots whose diameters were between 1 and 2 mm).

### 2.8. Analysis of the Effects of Water Stress, Potassium Deficiency, and Their Combined Effects

The effects of water stress and K deficiency, and their interactions, were calculated using the method proposed in Reference [43]. The effects of water deficiency on a given parameter were calculated from the non-limited K treatment plants (K2), as the difference between the W+ and W− plants, relatively to the unstressed plants (K2W+) as:

$$\text{Effects of W stress} = (\text{K2W}- - \text{K2W}+)/\text{K2W}+ \tag{7}$$

Similarly, the effects of K and, finally, the combined effects of W and K, can be written, respectively, as:

$$\text{Effects of K deficiency} = (\text{K0W}+ - \text{K2W}+)/\text{K2W}+ \tag{8}$$

$$\text{Combined effects of K and W stress} = (\text{K0W}- - \text{K2W}+)/\text{K2W}+ \tag{9}$$

The interactions between water stress and K deficiency were calculated as the difference between the observed combined effects of W and K and the sum of the effects of K and W taken individually, as described in Reference [44].

### 2.9. Statistical Analysis

Statistical analysis was performed using R software (R Development Core Team, Vienna, Austria [45]) to compute mean values and standard errors, and to test differences between treatments. Data were tested for normality (Shapiro's test) and for homogeneity of variance (Levene's test). Post-hoc differences in means were tested using Tukey's test. A linear model was used to analyze differences between each treatment, and a two-way analysis of variance (ANOVA) was performed to test for the effects of K and W levels and their interactions (W × K), based on a completely randomized system. We performed non-parametric test permutations (*n* = 999) to compare the mean values with $p \leq 0.05$ on root analysis (root area, specific root area, and root-to-leaf area index), since this test was more appropriate for small sample sizes.

## 3. Results

### 3.1. Plant K Status

Delaying the water stress treatment maintained similar ($p > 0.5$) plant K concentration at harvest (65 DAE) between the two water treatments (3.9 vs. 3.5 g kg$^{-1}$ of K in K0 treatment for W$-$ and W+ plants, respectively; and 10.2 vs. 8.9 g kg$^{-1}$ of K in K2 treatment for W$-$ and W+ plants, respectively). When expressed as a function of shoot water content (in mmol L$^{-1}$ of K), which might be more relevant when investigating the effects of K concentration on plant physiology, those values varied between 29 and 22 mmol L$^{-1}$ of K in the K0 treatment, and between 91 and 75 mmol L$^{-1}$ of K in the K2 treatment for W$-$ and W+ plants, respectively.

### 3.2. Plant Water Status

The volumetric soil water content of W$-$ plants dropped (Figure S1) just after water limitation treatment was applied. The decrease was faster for K1 and K2 treatment plants, as they asymptotically reached the threshold of 6% of volumetric water content at 500 dd (degree-days), whereas it took more than 100 dd for K0-treatment plants to reach the same value. For W+ plants, the volumetric water content never decreased below 16% for K2-treatment plants, and remained above 18% for K1- and K0-treatment plants. Final (700 dd) pre-dawn water potentials of W$-$ plants were $-0.32$ MPa, $-0.31$ MPa, and $-0.4$ MPa for K0, K1, and K2 plants, respectively. For W+ plants, these values only reached $-0.16$ MPa, $-0.12$ MPa, and $-0.17$ MPa for K0, K1, and K2 treatment plants, respectively.

### 3.3. Plant Development

Root development, characterized by the number of visible underground phytomers, was neither affected by W nor by K ($p > 0.05$). However, above-ground plant development, which was defined by the number of visible leaves as a function of thermal time (degree-days, dd) was affected by both K fertilization and water deficit (Figure 1, Table 1). The very first measurements showed that the increase in the number of leaves with dd was slower ($p \leq 0.001$) in plants grown with low K than in plants grown with high K. The gap became even larger when water stress was applied, i.e., after 380 dd (Figure 1). At the end of the experiment (700 dd), the combined K $\times$ W effects accounted for more than four extra leaves between the K0W$-$ and K2W+ treatments.

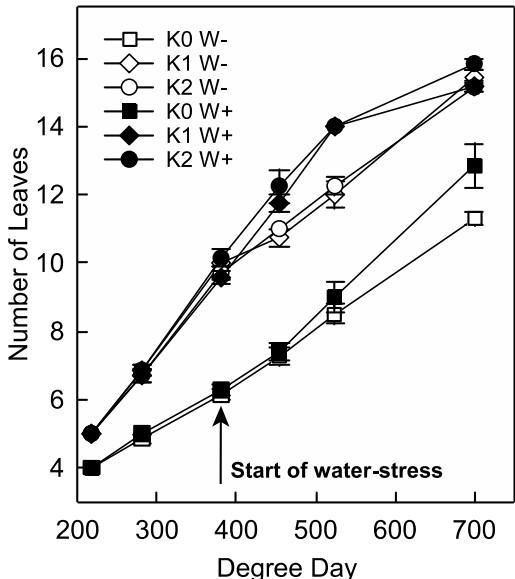

**Figure 1.** Number of leaves in relation to degree-days for well-watered (W+, black symbols) and water-stressed (W$-$, white symbols) corn plants ($n = 7$ plants; error bars are standard errors) for three potassium levels (K0 = low, K1 = normal, K2 = high).

**Table 1.** Analysis of variance probability values for water (W) and potassium (K) fertilization treatments and their interactions (W × K) on corn (*Zea mays* L.) morphological characteristics and on water transpiration and use efficiency.

| Statistical Tests | Parameters | *p* Values | | |
|---|---|---|---|---|
| | | **W** | **K** | **W × K** |
| Parametric | Number of Leaves | * | *** | * |
| | Leaf Area ($m^2$) | *** | *** | * |
| | Leaf Biomass (g) | *** | *** | *** |
| | Root Biomass (g) | ns | *** | ns |
| | LER (cm $dd^{-1}$) | *** | *** | ns |
| | Relative Senescent Leaf (%) | ns | *** | ns |
| | Phyllochron (dd) | *** | ** | ** |
| | SLA ($m^2\ kg^{-1}$) | ns | ns | ns |
| | R/(R + S) | ** | . | ns |
| | Transp. (L $plant^{-1}$) | *** | *** | *** |
| | Transp. per unit time and leaf surf. (L $cm^{-2}\ day^{-1}$) | ** | ns | ns |
| | Water Use Efficiency (g $L^{-1}$) | *** | ** | ** |
| Non-parametric | Root Area ($m^2$) | ns | ** | |
| | Specific Root Area ($m^2\ kg^{-1}$) | ns | ns | |

$p < 0.1$; * $p \leq 0.05$; ** $p \leq 0.01$; *** $p \leq 0.001$; ns: not significant; blank: not enough data. LER: leaf elongation rate, SLA: specific leaf area, R/(R + S): root to root plus shoot ratio, Transp.: transpiration. Number of leaves was taken at 700 degree days (dd).

For K1 and K2 treatments, the phyllochron was significantly lower ($p \leq 0.001$) on well-watered plants (34 dd) than it was on water-stressed plants (58 dd) (Figure 1, Table 1). Conversely, for the K0 treatment, phyllochrons were not significantly dependent on water treatments (58 and 60 dd for W+ and W−, respectively). When K1 W+ and K2 W+ plants were taken as reference, the effect of water stress had the same relative effects as those of K starvation, i.e., 24 more dd between the emergences of two leaves.

### 3.4. Leaf and Root Growth

At the end of the experiment (29 days after water stress was applied), the green leaf area appeared to be strongly influenced by K fertilization ($p \leq 0.001$), with values 4 to 5 times greater for K2 plants than for K0 plants (from 0.13 $m^2$ to 0.47 $m^2$ for W+ plants and from 0.07 $m^2$ to 0.37 $m^2$ for W− plants, respectively, Figure 2a). Water stress restricted leaf area ($p \leq 0.001$) by a constant value of 0.1 $m^2$ per plant, which corresponded to a reduction of 50%, 30%, and 25% for K0, K1, and K2 treatments, respectively. Differences in leaf biomass due to K fertilization were even greater ($p \leq 0.001$), showing an order of magnitude difference between K0 and K2 treatments (from 9 g to 86 g of leaf by plant for W+ treatment, and from 5 g to 54 g for W− treatment, Figure 2b). At a given K-level, water stress suppressed the leaf biomass by 30–40% (Figure 2b).

Root area was similar between water treatments ($p = 0.83$), but increased significantly ($p \leq 0.01$) with K fertilization from 0.1 $m^2$ (K0) to 0.7 $m^2$ (K2) (Figure 2c). Similarly, the changes in root biomass due to K fertilization ($p \leq 0.001$) were not affected by soil water content, increasing from 2 g in the K0 plants to 35–40 g in the K2 plants (Figure 2d). There were no significant differences ($p = 0.92$) in root biomass between water treatments.

The leaf elongation rate (LER) for leaf 10 (leaf taken as reference) was at its maximum in K1W+ and K2W+ treatments, with a similar value of 0.52 cm $dd^{-1}$ during the linear growing phase (Figure 3a). LER for K0W+ was 0.44 cm $dd^{-1}$, which represented a 15% reduction when compared to the maximum. This relative decrease was the same ($p = 0.29$) between K0 and K2 treatments under water stress (0.31 cm $dd^{-1}$ vs. 0.36 cm $dd^{-1}$ for K0 and K2, respectively). Regardless of the K treatments, water stress lowered LER by 30% ($p \leq 0.001$).

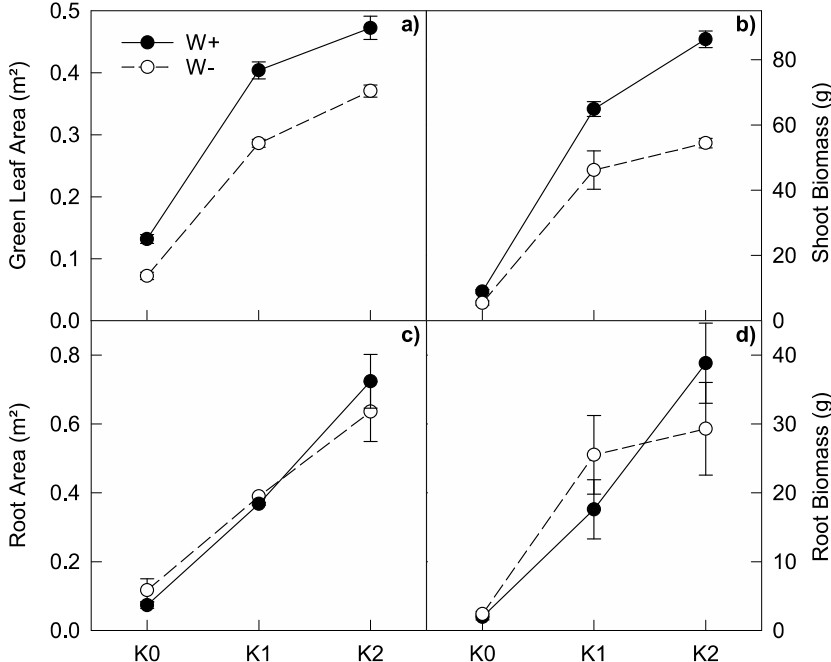

**Figure 2.** (**a**) Leaf area, (**b**) shoot biomass, (**c**) root area, and (**d**) root biomass of well-watered (W+, filled symbols) and water-stressed corn plants (W−, open symbols) as a function of potassium levels (K0 = low, K1 = normal, K2 = high) 29 days after the onset of the water stress (mean ± Standard Error; *n* = 7). Statistics are summarized in Table 1.

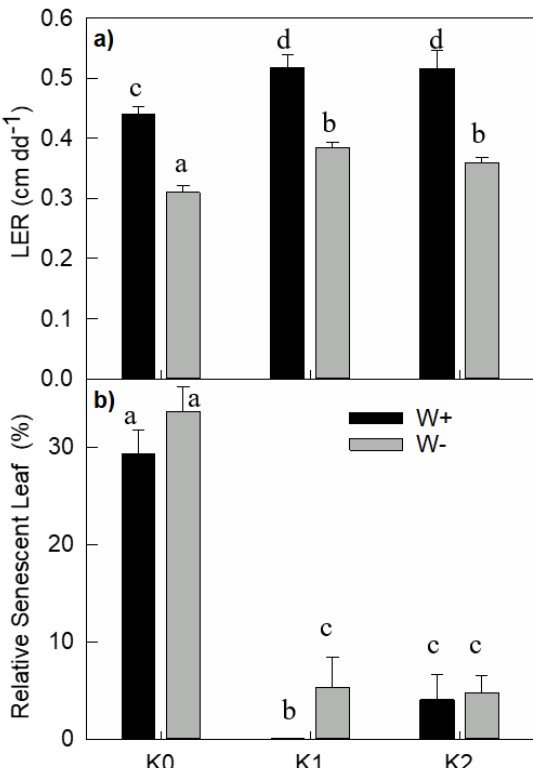

**Figure 3.** (**a**) Leaf elongation rate for leaf number 10 and (**b**) percent senescent leaf (relative to the total number of leaves) at the same development stage (10 visible leaves) of well-watered (W+, black bars) and water stressed (W−, grey bars) of corn plants as a function of three potassium levels (K0 = low, K1 = normal, K2 = high; mean ± Standard Error; *n* = 7). Senescence of K1W+ plants was equal to 0. Statistics are summarized in Table 1. Letters represent significant differences at 5% level.

### 3.5. Leaf Senescence

At the end of the experiment (700 dd), the proportion of senescent leaf area in the K0 plants reached 29% and 34% for W+ and W− treatments, respectively, whereas for K1 and K2 plants, it represented only a very small amount relative to well-watered and fertilized plants (≈5%). However, at that date, the developmental stages were heterogeneous (Figure 1). When we compared the proportion of senescent leaves at a given development stage (i.e., 10 visible leaves, the final stage for K0W− plants), the gap was even larger, as only the smaller bottom leaves of K1 and K2 plants began to dry (Figure 3b). The senescent leaf area was mainly influenced by K starvation, but the effect of water stress was not significant ($p = 0.13$). Finally, the effects of K nutrition on leaf development, LER and leaf senescence accounted for the observed differences in green leaf area (Figure 2a).

### 3.6. Morphology and Architecture at Organ and Plant Scales

Specific leaf area (SLA) of W+ and W− plants (18 m$^2$ kg$^{-1}$ and 16.5 m$^2$ kg$^{-1}$, respectively) did not significantly differ from each other ($p = 0.16$, Table 1), and there was no K effect on SLA (17.1 m$^2$ kg$^{-1}$, 16.6 m$^2$ kg$^{-1}$, and 18.3 m$^2$ kg$^{-1}$ for K0, K1, and K2 treatments, respectively, $p = 0.39$). In roots, the values of SRA showed no significant differences ($p = 0.21$) between water treatments, with overall mean values of 93 m$^2$ kg$^{-1}$ and 70 m$^2$ kg$^{-1}$ for W− and W+ treatments, respectively. There was a marginal effect of K fertilization on SRA ($p = 0.064$), with means of 112, 64, and 66 m$^2$ kg$^{-1}$ for K0, K1, and K2 treatments, respectively. However, there was a strong effect of K treatment on root architecture ($p \leq 0.001$), as revealed by the number of phytomer-associated primary roots, limited to 4 on K0 plants, but attaining 10 for other K treatments. As a consequence, the proportion of root area carried by each phytomer increased from phytomer 3 to phytomer 5, with the exception of K0W− plants, where the older roots accounted for most of the root area (Figure 4). This difference could be explained by a lower root number, as previously shown before, or by a difference in root morphology of the K-starved plants ($p \leq 0.01$).

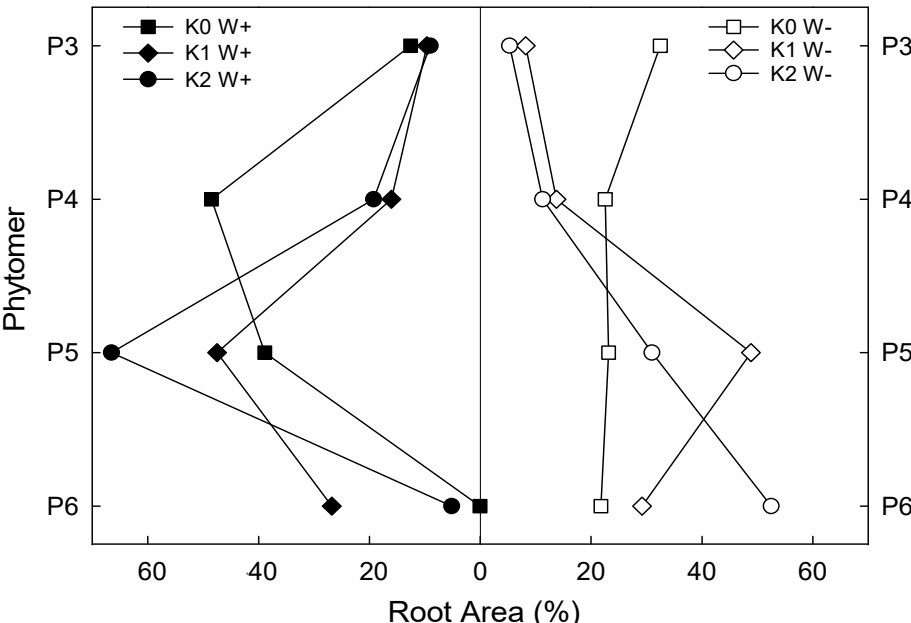

**Figure 4.** Distribution of root area (%) by phytomer for well-watered (W+, black symbols) and water stressed (W−, white symbols) corn plants grown at three different levels of potassium (K0 = low, K1 = normal, K2 = high; mean; *n* = 2). A phytomer consists in a repetition of constructional units represented by successive horizontal circles from which roots emerge at the base of the stem (labelled from P1 at the bottom to P6 at the top).

　　　The analysis of root morphology globally showed that the roots of the smallest class (<1 mm) accounted for 80% of the total root area (Figure 5a,b). Compared to K1 and K2 treatments, the proportion of roots comprised between 1–2 mm was higher on the K0-treatment plants. Taken as a whole, the roots of the smallest class (<1 mm) had a different pattern according to the particular water treatment. For W− plants, there was no effect of the K-treatment on the proportion of roots in this class. Conversely, the proportion of roots for the W+ plants increased with K nutrition from 65% to 80%. For the 1 mm-diameter class (see insert to Figure 5b), the higher proportion of roots occurred between 0.1 to 0.2 mm, regardless of the water treatment. The K0 plants had a slightly higher proportion of roots in the smallest diameter classes (0.1, 0.2, and 0.3 mm), than in the largest classes (0.4–1 mm).

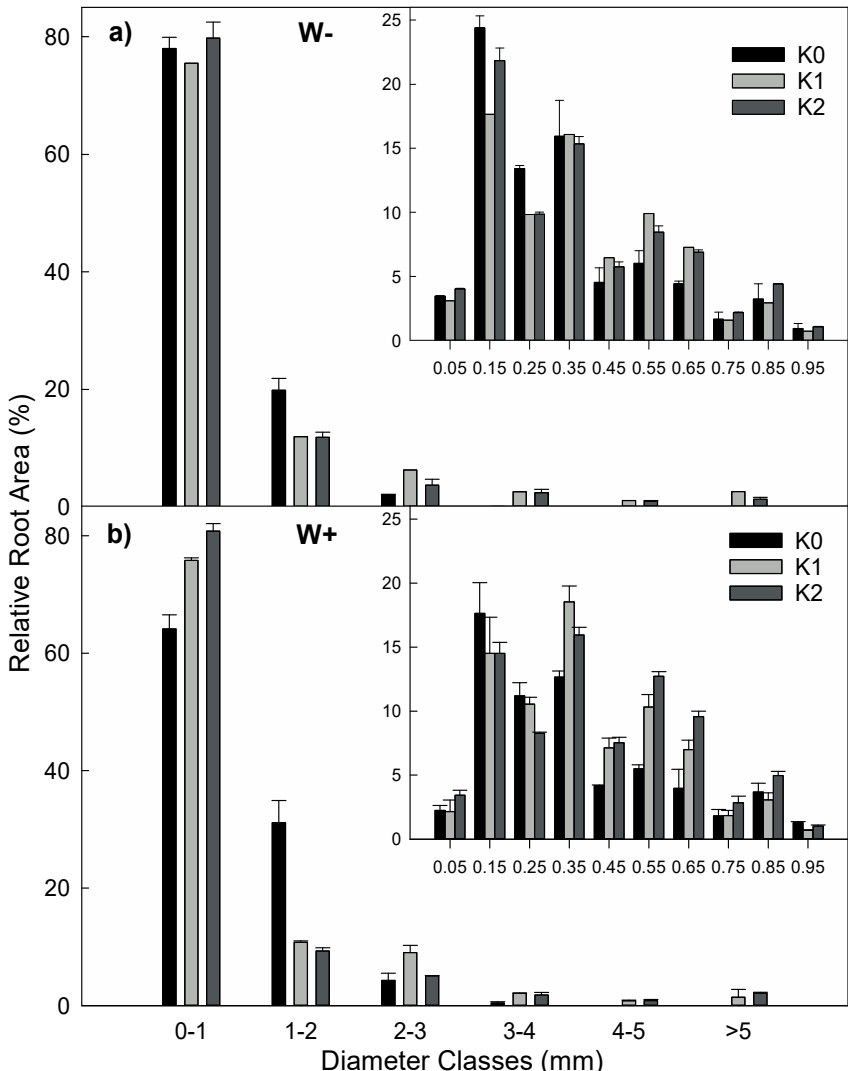

**Figure 5.** Distribution of relative root area (%) by root diameter classes (mm) for (**a**) well-watered (W+) and (**b**) water-stressed (W−) corn plants growing under low potassium (K0), normal potassium (K1), and high potassium (K2) levels (mean ± Standard Error; *n* = 2); insert represents the detailed distribution of relative root area between 0 to 1 mm.

　　　At the whole-plant level, the ratio of root dry biomass (R) to root plus shoot (S) dry biomass (R/(R + S)), increased significantly ($p \leq 0.01$, Table 1) under water stress from 0.23 to 0.33. This indicated that, in comparison to leaf growth, roots subjected to W− treatment grew more than those subjected to W+ treatment. There was a small K effect on the R/(R + S) ($p \leq 0.1$, Table 1), in which K starvation led to a lower allocation of dry matter to roots (0.24, 0.28, and 0.33 for K0, K1, and K2, respectively).

### 3.7. Effects of Water and K Treatments on Water Flux and Use Efficiency

The cumulated water transpired at the pot scale was measured from 380 to 700 dd. Results of transpiration spread over a large gradient, with 0.96, 6.83, and 7.12 L for K0, K1, and K2 treatment plants of the W− treatment, respectively. The values were nearly twice as high for the W+ plants, with 1.8, 10.3, and 12.8 L for the K0, K1, and K2 plants, respectively. K and W treatments had significant effects (Table 1). If we were to plot these values towards the final green leaf areas, we would obtain a linear regression ($r^2 = 0.90$) with K treatments from either side of the line; this means that K nutrition was not improving the fit of the model. The water transpired per unit time and per unit green leaf area, between two irrigation dates, was only water-treatment dependent (25% reduction in mean, $p = 0.0016$, Table 1, Figure 6a). Water-stressed plants lost on average 35 mL per day and per m$^2$ of green leaves, whereas the transpiration rose to a maximum of 49 mL for the non-stressed plants. Identically, water use efficiency (WUE), calculated from the beginning of water stress to the end of the experimentation, appeared to be both W and K dependent ($p = 0.03$ for K and <0.001 for W). The WUE for W+ and W− plants was 3.8 and 4.8 g L$^{-1}$, respectively. In detail, the water stress increased the values of WUE by 36%, 9%, and 27% of K0, K1, and K2 plants, respectively (Table 1, Figure 6b). The effect of the K treatment was not linear with the level of nutrition: K0 nutrition increased the average value of WUE compared to K1 and K2 treatments. On the contrary, the WUE value of K1 treatment was significantly lower than that of K2.

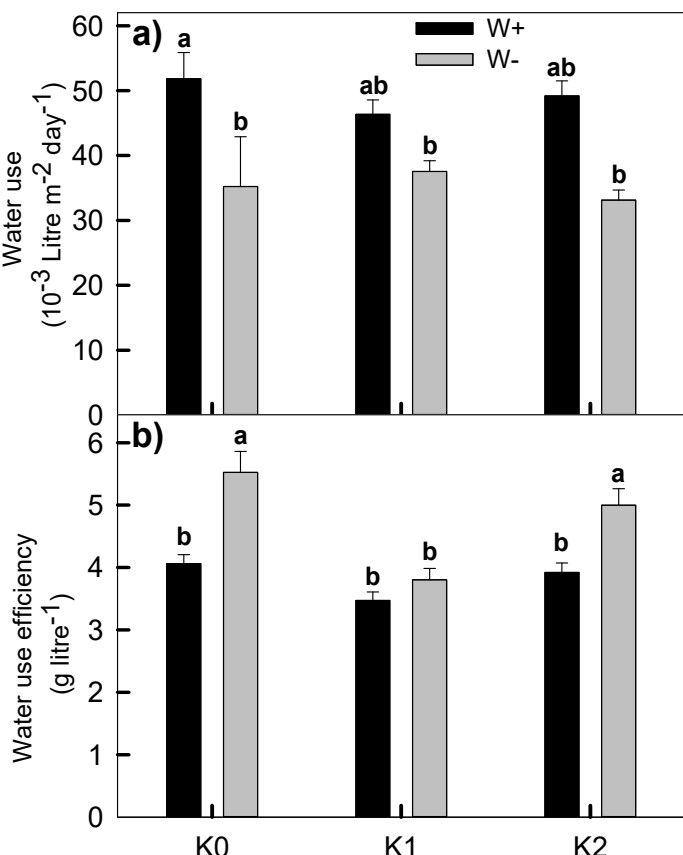

**Figure 6.** (**a**) Transpiration per unit time and per unit leaf area, and (**b**) water use efficiency calculated for the period with corn plants subjected to water shortage of well-watered (W+, black bars) and water stressed (W−, grey bars) as a function of three potassium levels (K0 = low, K1 = normal, K2 = high; mean ± Standard Error; *n* = 7). Statistics are summarized in Table 1. Letters represent significant differences at a 5% level.

*3.8. Water and Potassium Interactions on Morphological Traits*

When plants were suffering from combined water and K stresses, the global impact was always lower than what might have been expected if water effects alone had been added to K effects alone (Figure 7). This interactive effect, called an "attenuation effect," consisted in a partial or total dissimulation effect of one factor by the other. In decreasing order, this attenuation effect concerned the following plant traits: R/(R + S), shoot biomass, root biomass, and root area (Figure 6). A weak attenuation effect was also observed on leaf area (8.5%) and on the leaf elongation rate (5.8%).

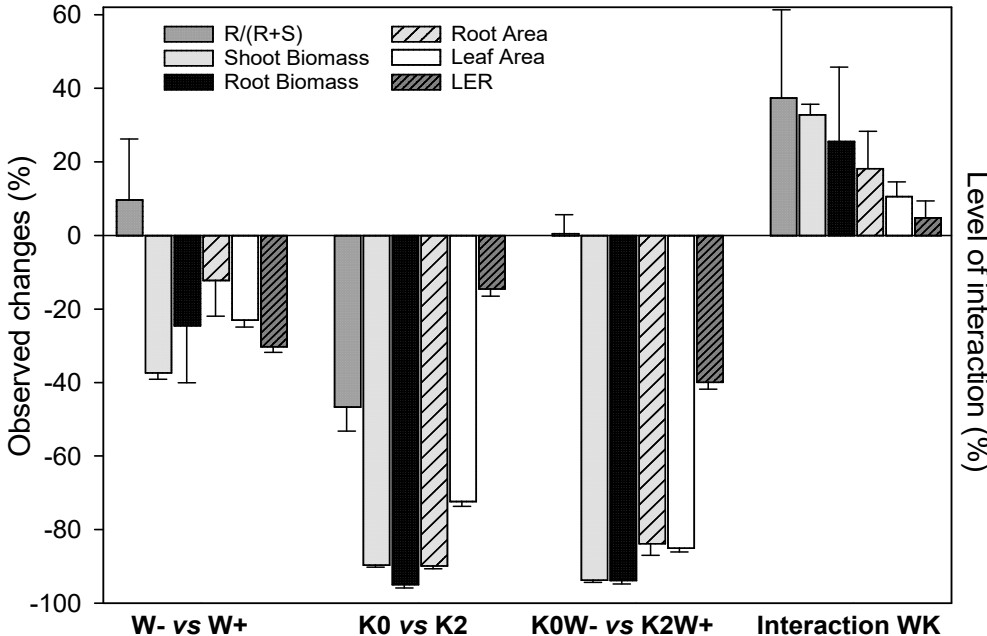

**Figure 7.** Relative variations in corn main morphological traits in response to water stress (W− vs. W+), to potassium deficiency (K0 vs. K2), or to both K and W constraints (K0W− vs. K2W+) (mean ± Standard Error). Values are obtained by calculating the ratio of the differences in growth or development to the non-limited treatment (K2W+). The interactions between water stress and K deficiency were calculated as the difference between the observed combined effects of W and K and the sum of the effects of K and W taken individually, as described in Reference [44].

## 4. Discussion

*4.1. Water and Potassium Status in Plants*

Water shortage, potassium (K) starvation, and their interactions on corn morphological and structural responses were observed in this study. K starvation was applied to the seedlings using soils with a very low K concentration (4 mg kg$^{-1}$ of exchangeable K, or 0.01 cmol kg$^{-1}$), unlike other studies dealing with fertilization experiments [18,44,46]. The applied water stress was also quite severe, but its application in the present study was progressive to prevent any mineral uptake disorders during early plant development. Early water shortage can deeply impact nutrient uptake, leading to mineral deficiencies [47]. Plants endured water stress for 29 days before sampling, with soil water content being maintained near the soil wilting point.

*4.2. Experimental Biases*

Three experimental biases were identified: two linked to the development rates of plants that were quite different between the most favorable (W + K2) and the most unfavorable (W − K0) treatments, and one linked to the growing medium. First, we could not prevent K0 plants from losing water more slowly than the others. As a consequence, the water stress was more rapidly reached on K1 and K4

plants of the W− treatment than on K0 plants. One solution would have been to use a hydroponics solution, with Polyethylene glycol (PEG) simulating water stress through osmotic stress [38]. In such conditions, plants suffer from the same osmotic stress, whatever their water consumption. We preferred to accommodate for the bias of different plant development rather than running the risk of having uncontrolled secondary growing effects. Previous experiments using PEG showed some difficulties with mimicking a gradual water stress. Second, the difference in plant development may have also interfered with some physiological measurements. To eliminate such a problem, we took certain precautions, i.e., focusing our LER measurements on the same leaf rank or expressing the senescent leaves relatively to total leaf areas. However, for WUE calculation, which was based on a long-term period (380–700 dd), the gap of four leaves between K0W− plants and others may have jeopardized the comparison of WUE values. Interactions between corn development, water stress, and WUE have been established [47]. Third, regarding root morphology, our experimental conditions did not allow us to measure root hair density, which was, however, assumed to be K-sensitive. In their experiments based on an aeroponic system, Hogh-Jensen and Pedersen [34] observed some interesting results on several species. In our case, the lack of significant tendencies on root systems between treatments may be explained by our observations focusing only on primary and lateral roots.

### 4.3. Contrasted Contributions of Water and K on Corn Development, Growth, and Water Use

At the shoot scale, the components of corn biomass and leaf area (plant development, leaf elongation rate, final leaf size, and leaf senescence) were all impacted by water and K stress, but in different proportions. For the leaf elongation rate and leaf area, K and water supplies constituted additive limiting factors, which can be interpreted as a lack of interaction from a statistical point of view. Hence, leaf growth is influenced by independent physiological mechanisms, which might be both potassium- and water-dependent. Sugar translocation to leaf meristems, which triggers cell division, is potassium-dependent [20,30,48], whereas turgor pressure, which influences cell elongation, is water-dependent [20]. Conversely, at the whole-plant scale, K deficiency, alone, globally explained the combined effects observed on shoot and root biomass. In other words, the expected damage to plant growth was attenuated, which represented, from a statistical point of view, a negative interaction. The dominance of one single limiting factor (K nutrition) can also be explained by the fact that K and water share most of the key physiological functions, such as turgor maintenance and water uptake, thereby influencing growth [23,49]. Even senescence, already known to be water dependent [13], was more sensitive to K stress than to water stress, when expressed at the same development stage (i.e., 10th visible leaf). We hypothesize that the lack of K impaired the neutralization of "reactive oxygen species", which are responsible for oxidative stress as mentioned in Reference [27], which in turn accelerated leaf senescence.

Regarding the root compartment, the decreased R/(R + S) ratio in K-deprived plants indicated that root growth was more severely reduced than shoot growth, which is a common trait of this mineral deficiency [50–52]. This can be explained, from a physiological point of view, by the lack of carbohydrate transport from C sources to sinks, as already stressed in other K-deficient studies for well-watered plants [53–55]. The low level of primary roots emergence on K0 plants mainly accounted for this difference, at least far more than any differences in root size distribution. These results indicated that K starvation jeopardized the positive effect of water stress on root growth (Figure 6). Under field conditions, this limitation is expected to reduce the amount of water transpired by plants and the subsequent nutrient uptake.

Water transpiration was mostly influenced by K through its impact on leaf area. Unexpectedly [2,28,29,56], there was no specific effect of K nutrition on stomatal regulation. Such a regulation was expected to partially offset the water losses. Transpiration rate per leaf surface was not K-dependent, even though it was measured over the long-term (400–700 dd). Instantaneous stomata conductance measurements (using a gas exchange analyzer) confirmed this absence of trend (data not shown). The WUE measured on a long-term basis was in the normal range of values (around

4 g L$^{-1}$) obtained in similar growing conditions [57,58], which reinforces, a posteriori, the rightness of our calculation method. As expected, water stress induced significantly higher values of WUE [22,59]. The K response was however much more surprising, going against many of the results obtained before [20,60,61]. However, we might attribute the significantly higher value of K0 plants to the delay of their development stage. K1 and K2 plants showed a more "logical" ranking, with an advantage for K2 plants, which benefitted from 20% more water efficiency compared to K1 plants, for W− treatment.

### 4.4. Finally, What Did We Learn about the Effects of K Nutrition on Drought Resistance?

K-nutrition is generally considered to alleviate water stress through the following mechanisms: better root soil-prospection, longer leaf lifespan, better cell growing capacity, and, above all, better osmotic regulation [60]. Previous experiments have shown such compensatory effects between K fertilization and water stress [27,49,62–64]. The relative contribution of each mechanism, which depends on the intensity of the K and W stresses and on their timing, is probably species-dependent. The actions of K nutrition on water-stress resistance were as follows: (i) K nutrition (K1, K2 vs. K0) enhanced root biomass, especially primary root growth. However, the diameter of laterals, which are mostly responsible for increased soil prospection, was not improved by K, at least not under water stress; (ii) the leaf lifespan was improved by K nutrition, although there was no advantage to over-fertilizing (K2) water-stressed plants with K. The lack of K was more deleterious than the lack of water; and (iii) K clearly promoted leaf growth, which in turn increased whole-plant water losses. The transpiration rate was, here, solely dependent on the plant leaf area and did not seem to benefit from any improvement in stomata or mesophyll conductance. This finding is, however, not corroborated by other recent results which particularly showed an influence on stomatal or mesophyll conductance [25,28,56]. Water use efficiency results did not show any advantage in applying more K than strictly needed for growth needs, but, as previously mentioned, the experimental bias linked to the delay in plant development did not allow us to form a conclusion. Hence, although K nutrition did partially compensate for water shortage in terms of leaf area and biomass, this may represent a real risk in terms of water mass balance whenever irrigation is not sufficient [65]. Our overall findings suggest that there was no significant advantage to over-fertilizing water-stressed plants with K. The highest level of K fertilization applied to the water-stressed plants did not allow the main plant growth components (green leaf area, leaf lifespan, root biomass, and shoot biomass) to be as high as the well-watered plants fertilized with a non-excessive quantity of K. From a practical point of view, we recommend that farmers should identify their soil-limiting parameters by obtaining soil analysis, and by adapting mineral inputs to the irrigation water facilities. Potassium, but also nitrogen or phosphorus, should be adapted to a realistic yield target. Over-fertilization can compromise the crop over the long term by enhancing early-stage water transpiration to the detriment of subsequent development stages (flowering, seed filling). As potassium is the only major mineral whose deficit decreases root growth, it is therefore crucial to give farmers judicious advice about their crop fertilization requirements. This advice should be based on the real quantity of K taken up by the plant, in order to prevent either extremely weak root growth or excessive shoot expansion.

**Supplementary Materials:** The following are available online at http://www.mdpi.com/2077-0472/8/11/180/s1, Figure S1: Evolution of (a) the volumetric Soil Water Content and (b) the corresponding soil water potential. The soil water content was obtained by weighing pots regularly (Equation (3)). The soil water potential was obtained by pre-dawn water potentials measurements at the end of the experiment as well as by a relationship obtained between soil water content and soil water potential on a previous greenhouse-pot experiment using the same soil.

**Author Contributions:** L.J.-M., J.-C.D., and Y.B. conceived and designed the experiments; L.J.-M., Y.B., J.L., C.H.A.-J., and J.-C.D. performed the experiments; L.J.-M., E.M., and J.-C.D. analyzed the data; L.J.-M. and E.M. wrote the paper.

**Funding:** This research received no external funding.

**Acknowledgments:** The authors would like to thank the COFECUB (n° Uc Sv 134/12) and USP/COFECUB (project 2011-25) program for allowing the mobility of French and Brazilian researchers. We are grateful to Mark

Bakker (INRA) for his comments on the original draft preparation and to Jean-Pierre Da Costa (Bordeaux Sciences Agro) for his advices on statistical analysis.

**Conflicts of Interest:** The authors declare no conflict of interest.

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
