# Peer review of "How Does Water-Stressed Corn Respond to Potassium Nutrition? A Shoot-Root Scale Approach Study under Controlled Conditions"

_agriculture, doi:10.3390/agriculture8110180_

Round 1
Reviewer 1 Report
The paper is interesting and adequate to the scope of the journal. Therefore I suggest to do some minor revisions prior to publication.
The statistics for several figures are presented in a separate table. It would improve the legibility of the paper to include each statistic (as asterisks) over the corresponding bar, as presented in most papers, to make the figures self-explanatory.
Figure 7 is presented without statistics. Although the data comes from operations with other data, the error should also be included in the calculations and in the figure. There are several mathematics methods to operate with errors.
Author Response
We agree with R1 with the fact that some figures could be improved by adding letters that indicate which treatments significantly differ form the others. We have added such letters on the 2 main histogram figures (3 and 6). The test have been made for all pairs, and the level of significance have been chosen at 0.05. The titles of the figures have been adapted.
We have added error bars on the values of the figure 7 (Standard Errors). We calculated the mean value of the reference treatment, and then calculated 7 relative calculations, which in turn let us obtained the S.E. The calculation of the S.E. of the Interaction, based on the Equation
Interaction = A - B – C supposes that :
Variance (Interaction) = Variance A + Variance B + Variance C
Then , we deduced the S.E. from the variance.
Reviewer 2 Report
The topic falls in the scope of “Agriculture” journal and the authors have worked on very important question. The study is of interest due to both scientific and technical reasons. The subject of study and the framework under which the study is developed are of outstanding importance in a global context due to the implication of the proposals of this study. Experiments seem well designed and methods used appropriated. Results are well described and Discussion based on the results and do not speculative. Some minor typos, grammar and syntax errors should be carefully revised and corrected accordingly. I suggest only the correction of a few minor issues (detailed below).
Line 238: Authors should indicate the units only according to the International System of Units (SI). The language-dependent terms “part per million”, “part per billion”, and “part per trillion”, and their respective abbreviations “ppm,” “ppb,” and “ppt” (and similar terms and abbreviations), are not acceptable for use with the SI to express the values of quantities.
Line 472: “…as described in Christina et al. (2015)” should be “…as described in Christina et al. [59]” or “…as described in [59].”
Author Response
The corrections suggested by the R2 have been done (see below) as well as a deep reviewing for typos, grammar and syntax errors.
All "s" in "Fertilisation" or "Fertilisers" have been changed to "z"
All "s" in analyse" have been changed to "z"
L40-42
Fertilisation and, in particular, potassium (K) application, is among the most commonly accepted strategies used to alleviate symptoms of water stress
→ Fertilisation with, in particular, the application of potassium (K), is one of the most commonly accepted strategies used to alleviate symptoms of water stress
L50
Cells guard are known to be responsive to K
→ Guard cells are known to be responsive to K
L53-54
Contradictory results have, however, been pointed
→ Contradictory results have, however, been pointed out
L55
Moreover, recent data clearly show that the potassium's influence on transpiration is linked to
→ Moreover, recent data clearly show that potassium's influence on transpiration is linked to
L62
K facilitates the transformation in leaves of glucose into sucrose which, thus favouring shoot-to-root sugar transport ,
→ K facilitates the transformation in leaves of glucose into sucrose, thus favouring shoot-to-root sugar transport
L103
Root development, characterised by the number of visible underground phytomers, was neither affected by W nor K
→ Root development, characterised by the number of visible underground phytomers, was neither affected by W nor by K
L161 added to the title of figure 3:
→"Letters represent significant differences at 5% level.
L239 added to the title of figure 6:
→"Letters represent significant differences at 5% level. "
L259 4 ppm of exchangeable K
→4 mg.kg-1 of exchangeable K
L273-274
We preferred to accommodate for the bias of different plant development rather than running the risk having uncontrolled secondary growing effects
→We preferred to accommodate for the bias of different plant development rather than running the risk of having uncontrolled secondary growing effects.
L281
Interactions between corn development, water stress and WUE has been established
→ Interactions between corn development, water stress and WUE have been established
L282
our experimental conditions did not allow us to measure root hair density, however assumed to be K-sensitive
→ our experimental conditions did not allow us to measure root hair density which was, however, assumed to be K-sensitive
L285-286
the lack of significant tendencies on root systems between treatments may be explained by our observations only focusing on primary and lateral roots
→ the lack of significant tendencies on root systems between treatments may be explained by our observations focusing only on primary and lateral roots.
L297-298
In other words, the expected damage on plant growth was attenuated, which represented, from a statistical point of view, a negative interaction
→ In other words, the expected damage to plant growth was attenuated, which represented, from a statistical point of view, a negative interaction
L309-310
at least much more than any differences in root size distribution
→ at least far more than any differences in root size distribution
L350-351
we recommend that farmers should identify their soil-limiting parameters by obtaining soil analysis, adapting mineral inputs
→ we recommend that farmers should identify their soil-limiting parameters by obtaining soil analysis, and by adapting mineral inputs
L385
there were 17 weighing
→ there were 17 weighings
L394
grown in the glass house
→ in the glasshouse
L397
each last ligulate leaf was carved in order to insert their main rib into the conical seal
→ each last ligulate leaf was carved in order to insert its main rib into the conical seal
L424
Moreover, we paid particular attention on leaf 10
→ Moreover, we paid particular attention to leaf 10
L429
Daily values of green leaf areas were needed to be put in parallel to water transpiration data. So we used a logistic curve fitted on the 6 measurement dates to make interpolations
→ Daily values of green leaf areas were needed in order to parallel water transpiration data, we used a logistic curve fitted on the 6 measurement dates to make interpolations.
L514-515
as described in Christina et al. (2015).
→ as described in [39]
L519
Statistical analyses
→ Statistical analysis